# Revealing Inflammatory Indications Induced by Titanium Alloy Wear Debris in Periprosthetic Tissue by Label-Free Correlative High-Resolution Ion, Electron and Optical Microspectroscopy

**DOI:** 10.3390/ma14113048

**Published:** 2021-06-03

**Authors:** Rok Podlipec, Esther Punzón-Quijorna, Luka Pirker, Mitja Kelemen, Primož Vavpetič, Rajko Kavalar, Gregor Hlawacek, Janez Štrancar, Primož Pelicon, Samo K. Fokter

**Affiliations:** 1Ion Beam Center, Helmholtz-Zentrum Dresden-Rossendorf e.V., Bautzner Landstrasse 400, 01328 Dresden, Germany; g.hlawacek@hzdr.de; 2Condensed Matter Physics Department, Jožef Stefan Institute, Jamova Cesta 39, 1000 Ljubljana, Slovenia; luka.pirker@ijs.si (L.P.); janez.strancar@ijs.si (J.Š.); 3Department of Low and Medium Energy Physics, Jožef Stefan Institute, Jamova Cesta 39, 1000 Ljubljana, Slovenia; mitja.kelemen@ijs.si (M.K.); primoz.vavpetic@ijs.si (P.V.); primoz.pelicon@ijs.si (P.P.); 4Department of Pathology, University Medical Centre Maribor, Ljubljanska ulica 5, 2000 Maribor, Slovenia; rajko.kavalar@ukc-mb.si; 5Department of Orthopaedics, University Medical Centre Maribor, Ljubljanska ulica 5, 2000 Maribor, Slovenia; samo.fokter@guest.arnes.si

**Keywords:** adverse local tissue reactions (ALTR), total hip arthroplasty (THA), titanium alloy, wear debris, periprosthetic tissue, correlative microscopy, micro-PIXE, SEM-EDS, HIM, hybrid confocal fluorescence and reflectance microscopy, fluorescence lifetime imaging microscopy (FLIM), fluorescence hyperspectral imaging (fHSI)

## Abstract

The metallic-associated adverse local tissue reactions (ALTR) and events accompanying worn-broken implant materials are still poorly understood on the subcellular and molecular level. Current immunohistochemical techniques lack spatial resolution and chemical sensitivity to investigate causal relations between material and biological response on submicron and even nanoscale. In our study, new insights of titanium alloy debris-tissue interaction were revealed by the implementation of label-free high-resolution correlative microscopy approaches. We have successfully characterized its chemical and biological impact on the periprosthetic tissue obtained at revision surgery of a fractured titanium-alloy modular neck of a patient with hip osteoarthritis. We applied a combination of photon, electron and ion beam micro-spectroscopy techniques, including hybrid optical fluorescence and reflectance micro-spectroscopy, scanning electron microscopy (SEM), Energy-dispersive X-ray Spectroscopy (EDS), helium ion microscopy (HIM) and micro-particle-induced X-ray emission (micro-PIXE). Micron-sized wear debris were found as the main cause of the tissue oxidative stress exhibited through lipopigments accumulation in the nearby lysosome. This may explain the indications of chronic inflammation from prior histologic examination. Furthermore, insights on extensive fretting and corrosion of the debris on nm scale and a quantitative measure of significant Al and V release into the tissue together with hydroxyapatite-like layer formation particularly bound to the regions with the highest Al content were revealed. The functional and structural information obtained at molecular and subcellular level contributes to a better understanding of the macroscopic inflammatory processes observed in the tissue level. The established label-free correlative microscopy approach can efficiently be adopted to study any other clinical cases related to ALTR.

## 1. Introduction

One of the major unresolved problems in joint replacement is the adverse biological reaction to implant materials that can activate the immune response and thus cause severe inflammation [1,2]. These processes start well before any evident symptoms. Therefore, it is important to develop methods for early identification and prevention of catastrophic failures. The failure of the implant represents the final stage of a series of successive events activating inflammation, which spreads from the sites with concentrated micro and nanoparticle debris and is accompanied by the generation of reactive oxygen species and other upregulated mediators causing periprosthetic osteolysis [3]. Furthermore, the inflammatory response of surrounding tissue can increase the nearby acidity [4], leading to an even more corrosive environment for metallic implants that can cause more release of metallic debris to the surrounding tissue and trigger a cell reaction cascade ending in implant rejection [5]. A better understanding of the etiology and pathophysiology of relevant biological and chemical principles is necessary for earlier diagnosis and treatment of the loosening implant [2].

In recent years, novel prostheses (femoral stems) with bi-modular junctions have been designed for total hip arthroplasty (THA). However, higher revision rates have been reported compared to traditional monoblock stems [6], possibly due to more pronounced neck–stem junction-related corrosion, deformation and mechanical failure, leading to the production and the release of metallic debris into the surrounding tissue [7,8,9]. Metallic-associated adverse local tissue reaction (ALTR) to metal debris presents a severe problem in selecting the materials for the THA engineering, as very little is known about the long-term biological effects of new biomaterials with designated better mechanical and preservation properties. Metal ions and nanoparticles, such as aluminium (Al), chromium (Cr), cobalt (Co) and nickel (Ni) have been shown to potentially cause severe adverse, clinically relevant toxic effects by binding to cellular proteins and enzymes modulating cytokine expression [10] and initiating a cascade of phagocytosis related events leading to implant loosening [11]. Besides, titanium (Ti) particles were also found to elevate cytokine expression resulting in the inflammatory response and periprosthetic bone loss [12], where the material degradation is mostly confined to the junction site of the bimodular THA [13]. Moreover, the release of metallic ions in the bloodstream, which has been measured in several patients with potential THA failure [14], can lead to toxic effects as well, especially in elderly patients due to a dysregulated metal homeostasis [15].

Current state-of-the-art imaging techniques in clinics to gather structural information of the post-implant effects on the surrounding tissue and biochemical information of cytokines involved in the immune response rely on magnetic resonance imaging (MRI) [16,17,18], computed tomography (CT, SPECT-CT [19]) and immunohistochemical labelling [10,20], respectively. Despite the techniques being a gold-standard for ALTR diagnostics on a tissue scale, MRI and CT especially, have limited spatial resolution of typically a few hundred µm, thus not being able to provide structural and functional information of the debris effect on the surrounding environment on a single or further subcellular scale. Moreover, these imaging techniques lack the quantitative chemical information that can be acquired by alternative techniques based on focused electron or ion beams. While energy dispersive X-ray spectroscopy (EDS) coupled to a scanning electron microscope (SEM) can provide high lateral resolution of a few nm at the expense of limited elemental detection sensitivity of 0.01 wt% [21], particle-induced X-ray emission with focused high-energy proton beams (micro-PIXE), on the other hand, can provide high elemental detection sensitivity down to 0.0001 wt% [22] with the drawback of a limited lateral resolution to a few hundred nm [23]. The techniques differ significantly in the depth of analysed layer in the sample, with EDS providing elemental information from the depth of few tens of nanometres, whereas micro-PIXE from the depth of few tens of micrometres. The specific strengths of micro-PIXE and EDS can favourably be exploited for the elemental characterization of wear debris and were thus chosen in our workflow using the correlative microscopies approach.

Despite correlative microscopy (CM) being a relatively young field, recent developments in advanced microscopies, instrumentation and sample preparation have led to many successful applications that have provided insightful elemental, structural, biochemical or functional information on the measured complex biological samples [24,25,26]. We applied the CM approach to the periprosthetic tissues contaminated with metal debris in the vicinity of a THA with a suddenly fractured interchangeable neck to perform in-depth tissue analysis around the junction of the femoral stem and the modular neck. Still, a combination of different imaging and spectroscopy techniques, including histology, SEM-EDS and Fourier Transformed Infra-Red (FTIR), have recently been demonstrated as useful tools to image and determine the biochemical information of wear and corrosion in tissue samples [27]. Furthermore, by a combination of transmission electron microscopy (TEM-EDS), SEM-EDS and X-ray diffraction spectrometry (XRD), the size, shape, element composition and crystal structure of metal particles is correlated with histological features showing tissue infection [5]. This indicates how important the tissue characterization is on a nanoscale to better understand the immune response on larger scales.

In our study, we thus apply a CM approach for the characterization of wear debris and its impact on the surrounding periprosthetic tissue, collected during revision surgery of a THA with a bi-modular femoral stem. We demonstrate that a new label-free multimodal and multiscale experimental workflow (Figure 1) by combining different complementary imaging and spectroscopy techniques can provide unique and insightful structural and functional information on the studied tissue, covering wide spatial scales down to nm. We implemented: (i) multimodal optical microspectroscopy based on hybrid confocal laser scanning microscopy (CLSM), near infrared (NIR) reflectance confocal microscopy (RCM), fluorescence lifetime imaging (FLIM) and fluorescence hyperspectral imaging (fHSI) for wear debris impact on the surrounding molecular and cellular environment, exploiting the autofluorescence detection; (ii) helium ion microscopy (HIM) for nm resolution and surface sensitivity imaging of the debris size, shape and topography; (iii) SEM-EDS for elemental analysis and imaging of individual wear debris and (iv) micro-PIXE for more sensitive elemental analysis and quantification of the wear debris on a micron to mm-scale.

## 2. Materials and Methods

### 2.1. Preparation of Tissue Samples

The tissue samples were collected during the THA revision surgery from a patient who suffered from a suddenly fractured long straight modular neck of a commonly used bi-modular femoral stem (Profemur^®^ Z, Wright Medical Technology, now MicroPort Orthopedics, Arlington, TN, USA) with an oval Morse taper as the neck-stem junction. Both neck and stem were made of Ti alloy containing Ti, Al and Vanadium (V): 90Ti, 6Al and 4V (wt%). The study was approved by the author’s (SKF) institutional review board (IRB) and the patient gave informed consent to participate. Samples were sent for cultures, histologic and high-resolution multimodal examination. The biopsy tissue was stained to delimit the surgical biopsy borders with the Davidson Marking System^®^ (DMS) in blue colour (#3408-5), from Bradley Products, Inc. (Minneapolis, MN, USA). The tissue was fixed in formalin (10% neutral buffered formalin) and placed in paraffin blocks with the standard process using an ExcelsiorTM AS Tissue Processor (Thermo Fisher Scientific, Waltham, MA, USA) and Tissue Embedding System TES99 (MEDITE Cancer Diagnostics, Chicago, IL, USA) (Figure 2).

### 2.2. Hybrid Fluorescence Confocal Laser Scanning Microscopy (CLSM) and NIR Reflectance Confocal Microscopy (RCM)

Experiments were performed on a custom-built super-resolution stimulated emission depletion (STED) microscope (Abberior Instruments, Gottingen, Germany) combined with a tuneable two-photon excitation laser system (Chameleon Discovery, Coherent, Santa Clara, CA, USA). Briefly, imaging was done with Olympus 60× water immersion objective (NA = 1.2) using two pulsed lasers simultaneously with a fast gating system controlled by the FPGA unit. For detecting fluorescence of endogenous tissue fluorophores, a pulsed diode laser was used (λ = 561 nm, pulse length 120 ps and repetition rate 80 MHz). For experiments with reflected and scattered near-IR light, a tuneable pulsed near-IR laser was used (λ = 760–780 nm, pulse length 100 fs and repetition rate 80 MHz). Both fluorescence and reflected/scattered light were detected using avalanche photodiodes (APD, SPCM-AQRH, Excelitas, Mississauga, ON, Canada) with photon detection efficiency (PDE) above 50% at the whole detected visible spectrum. Fluorescence was detected within spectral window λ = 580–625 nm using dichroic and bandpass filter (both Semrock). For a detailed optical setup, see the schematics on Appendix A.

### 2.3. Fluorescence Lifetime (FLIM) and Fluorescence Hyperspectral Imaging (fHSI)

The microscope system was upgraded with 16-channel multiwavelength photomultiplier detectors PML-16 GaAsP and multidimensional TCSPC detection system (both Becker&Hickl, Berlin, Germany) to simultaneously detect signals of up to 16 spectral channels and the corresponding fluorescence lifetime decays [28]. For FLIM imaging, signals collected on 16 GaAsP detectors arriving in the same time interval were summed accordingly. FLIM decay curves obtained in each pixel of an image were analysed using SPCImage 7.3 software (Becker&Hickl, Berlin, Germany). Analysis of FLIM decay curves was done by two-component, double exponential fitting convoluted with instrument response function (IRF). Pixel exposure time and image binning were set to obtain a sufficient signal-to-noise ratio to eligibly fit the decay curves with double exponential fitting function. Fitted FLIM decay curves from each pixel were color-coded according to the average lifetime *τ*_m_ with the fastest decays shown in red and longer decays in blue.

In addition to FLIM, fHSI was performed by using 16 spectral channels in the wide spectral window from 560 nm to 760 nm, set by the diffraction grating position in the spectrograph. The high quantum efficiency of nearly 50% in practically the whole detected visual spectra was achieved by GaAsP photomultiplier tubes (PMT) detectors. All photon counts collected on TCSPC for each pixel were summed for each spectral channel and represented in the spectral curve. Hyperspectral data were fitted by a suitable empirical spectral model, log-normal function, as described before [29]. The developed model was shown to be capable of resolving as small as 1 nm changes in the spectra [30] with high numerical stability during optimization. Up to three spectral parameters λ_max_, w and a were fitted with an achievable peak position resolution of 1 nm. Image binning was used before spectral fitting to increase the sensitivity with the slight cost of image resolution.

### 2.4. Helium Ion Microscopy (HIM)

High-resolution images on nanoscale were acquired using a helium ion microscope (Orion NanoFab, Zeiss, Jena, Germany). This microscope, equipped with GFIS injection system and an additional time of flight backscattering spectrometry, can achieve lateral resolution down to 0.5 nm in the energy range of 10–35 keV He ions [31]. An embedded specimen of periprosthetic tissue was mounted on an HIM sample holder by gentle attachment using electric conductive carbon tape. Imaging was done by collecting secondary electrons (SE1) emitted from the first few nm of the sample under the following experimental parameters: helium ion energy (30 keV), ion current (1.7 pA), and chamber vacuum (3 × 10^−7^ hPa). Field of view of the acquired images varied from 1000 × 1000 µm^2^ down to 4 × 4 µm^2^ with a minimal pixel step size of 2 × 2 nm^2^. Imaging was performed both on a non-tilted and on a tilted sample stage (45 degrees) for better surface 3-D visualization.

### 2.5. Scanning Electron Microscopy with Energy-Dispersive X-ray Spectroscopy (SEM-EDS)

High-resolution imaging and chemical analysis took place in the FEI HeliosNanolab 650 scanning electron microscope (SEM) using energy-dispersive X-ray spectroscopy (EDS) (Hillsboro, OR, USA). All the samples were coated with an approximately 10 nm carbon layer to prevent the charging effects of a biological specimen during electron irradiation. The EDS spectra were collected with the following experimental parameters: electron acceleration voltage (15 kV), electron current (200 pA) and chamber vacuum (10^−6^ hPa). Under such conditions, we achieved an energy resolution of 145 eV at Mn Kα line and elemental sensitivity of approximately 0.01 wt% (X-Max SDD, Oxford Instruments, Abington, UK).

### 2.6. Particle Induced X-ray Emission (Micro-PIXE)

The tissue samples were prepared for micro-PIXE analysis by cutting thin tissue slices from the paraffin block with a microtome, with selected thicknesses ranging from 10 to 40 µm. The slices were sandwiched between two 1 µm thick mylar windows spanned over frames made of Aluminium. The mylar windows allow the stretched support of the thin slices with a very low effect on both the incoming proton beam and the X-rays coming out from the sample. Micro-PIXE measurements were done in a Microbeam vacuum chamber, with a 3 MeV proton beam with an approximate size of 1 × 1 µm^2^ and analytical ion currents up to 300 pA (Microanalytical Center, Jožef Stefan Institute [32]). Thanks to a dedicated multicusp ion source featuring high H− beam brightness [33], the dimensions of the 3 MeV proton beam could be reduced significantly, maintaining the same beam intensity to optimize both the lateral resolution and sensitivity of micro-PIXE analysis. The characteristic X-rays were detected by a pair of X-ray detectors, with a silicon drift (SDD) detector for soft X-rays (0.8 to 4 keV), and intrinsic germanium (iGe) detector for X-rays with higher energies (3 to 30 keV), respectively. Both X-ray detectors run simultaneously with an on–off axis scanning transmission ion microscope (STIM) to determine the proton exit energy after passing through the sample, providing the information on sample area density, which is used in the quantification of the elemental concentrations from the measured PIXE spectra. For a detailed instrumentation setup, see Appendix A.

## 3. Results and Discussion

### 3.1. Wear Debris Cause Oxidative Stress in the Nearby Biological Tissue Detected by Label-Free Multimodal Optical Microspectroscopy

To better understand the biological impact of wear debris on the surrounding periprosthetic tissue on a submicron scale, a set of label-free multimodal optical microscopies was implemented first. It is known that in particular fluorescence-based microscopies with high sensitivity and specificity can provide enough information to fully comprehend the properties of complex tissues [34]. Thus, we tracked the local fluorescence properties of endogenous fluorophores in the direct vicinity of the wear debris.

First, wear debris were measured by widefield microscopy (WFM) using light-emitting diode (LED) illumination (400 nm), multi-band emission window and an RGB camera on a 10× magnification setup (Figure 3A). Fluorescence contrast can be observed between blue, orange and darker regions revealing different tissue structures with different fluorescence spectra. According to previously observed histological observations [35], the structure with stripe-like regions shown with yellow/orange colour represents the inflamed pseudomembrane of the periprosthetic tissue. Darker spots of sizes up to 200 µm are most likely attributable to the metallic wear debris (metallosis) that are spread throughout the measured tissue sample.

To get a better picture of a wear debris size, shape and its impact on the surrounding biological tissue, the marked region was measured with hybrid fluorescence CLSM (λex = 561 nm) and NIR RCM (λex = 765 nm) using 60× magnification with NA = 1.2 (Figure 3B). The regions with an increased NIR reflection (pixels in red colour) are observed and correspond to highly refractive (crystal-like) surfaces. According to a recent study [36], the elevated NIR intensity in these regions is really due to the highly reflective nature of metal debris. The signal gathered through NIR reflection thus nicely shows the distribution and incorporation of small, up to 10 µm sized, metal debris throughout the tissue. Moreover, the nonhomogenous and locally elevated endogenous fluorescence signal (in green) was observed in the surrounding tissue, showing on the biological response to the nearby wear debris. The elevated fluorescence signal could be the first indication of the oxidative stress as a response to pathological conditions or toxic compounds [37] such as metal wear debris, just recently been found to induce the formation of reactive oxygen species [38]. Even more, in our latest study, metal oxide nanomaterial has also been found to be one of the key players in chronic inflammation [39].

The oxidative stress is typically expressed through the production of autofluorescent lipofuscin, being accumulated within lysosomes [40]. The measured bright fluorescent spots with a typical size of 1–2 µm could indeed present lysosomes rich in the autofluorescent age pigment lipofuscin [41]. To verify the presence of lipofuscin, optical FLIM and fHSI were applied (Figure 3C,D). Both techniques are capable of achieving high sensitivity to the local molecular properties by resolving fluorescence decays with a time resolution down to 100 ps [42] and resolving spectral changes with nm resolution [30]. By time resolved FLIM analysis, the identification and discrimination between different endogenous fluorophores and the wear debris itself was clearly seen (Figure 3C). Four distinct molecular regions were found according to the calculated weighted mean fluorescence lifetime: region 1 in red (τ_m_ = 0.4 ± 0.1 ns), region 2 in yellow (τ_m_ = 0.7 ± 0.1 ns), region 3 in green (τ_m_ = 1.1 ± 0.2 ns) and region 4 in blue (τ_m_ = 2 ± 0.2 ns). Figure 3B shows the complete spatial overlap of region 1 with the lowest τ_m_ and the wear debris sites measured with NIR RCM. The decrease in the fluorescence decay time on the wear debris sites is most likely caused by the interaction of the excited electrons of the biological molecules with the conducting metal surface in the direct vicinity of a few nm. The interaction mechanism has recently been described as the spectral overlap between the fluorescence emission and the localised surface plasmon (LSP) spectra [43]. FLIM was thus, for the first time, shown to be capable of indirectly finding and identifying the wear metal debris in the tissue, with high applicability potential.

The fastest decay time of the autofluorescent molecules surrounding wear debris was shown on the bright fluorescent spots in yellow (region 2), which could either be fluorophore aggregates or, more likely, accumulated molecules inside the cell compartments. Detailed FLIM analysis with two-component fitting revealed lifetimes and their relative amplitudes of τ_1_ = 0.45 ns (a_1_ = 85%) and τ_2_ = 1.7 ns (a_2_ = 15%) which are in fairly good agreement with the measured lifetime components of lipopigment lipofuscin [44,45]. To further prove the presence of lipofuscin, fHSI was applied, using a log-normal spectral fitting algorithm (Figure 3D). Bright spots (in red) are nicely distinguished from the rest of the endogenous fluorophores in the tissue (in green). The difference in the peak emission was Δλ = 10 nm, red-shifted on the bright spots (λ_max_ = 620 ± 2 nm). The spectral analysis showed a spectral width (FWHM) of approximately 100 nm (for details, see Appendix A), which is typical for lipopigments and lipofuscin [46].

Thus, these results support the previous hypothesis of the bright spots representing lysosomes rich with accumulated lipofuscin, which is more than likely a consequence of oxidative stress of the nearby wear debris. In addition, such small micron-sized wear debris has a much larger specific surface area compared to larger debris and thus can cause increased surface reactivity that can exhibit greater biological activity in the surrounding tissue [47].

### 3.2. Wear-Debris Nanotopography Detected by HIM Reveals Significant Fretting and Corrosion That Can Affect Biological Tissue

To uncover more detailed surface properties of the observed micron and sub-micron wear debris affecting the nearby biological environment, the same tissue specimen was transferred to HIM. The technique is capable of achieving sub-nm spatial resolution with superior surface sensitivity and depth-of-field over the scanning electron microscope (SEM). The biological samples for HIM do not require a surface conductive layer coating for charge compensation [48], thus preventing any potential modification or contamination of the surface, making HIM particularly well suited for the workflows in the correlative microscopy.

CM workflow, applied here, was done by widefield and the combination of hybrid fluorescence CLSM and NIR RCM to localize and identify wear debris and its impact on the surrounding tissue (Figure 4A), as explained in detail in the previous section. The same specimen was then transferred to HIM, where the same region was found and overlayed with a successful image registration (Correlia ImageJ plug-in [49]) via wear debris used as the fiducial points (Figure 4A, right). Detailed surface analysis on the wear debris was performed next. Due to the powerful focusing capability of He ion beams with the probe size below nm [50], the technique enabled high-resolution imaging to identify the morphology and topography of the wear debris down to the nm scale (Figure 4B), as was recently shown on nanoporous materials [51]. A broad distribution of the debris size, shape and morphology was observed on a wider field of view (on the right), whereas distinct surface nanomorphologies were only revealed at the more close-up image (on the left). High magnification revealed wear debris regions with a pronounced surface roughness of up to a few hundred nm and with an ultra-low surface roughness of only a few nm (see the white arrows). Furthermore, the high surface sensitivity and large depth of field of HIM [52] enlightened the nm topographical features of the crack made through the flat wear debris, indicating the brittleness of the metal alloy on such scales. The wide extent of morphologies can be attributed to the substantial wear damage due to fretting and corrosion that can induce an inflammatory effect on the surrounding biological environment [53]. The topographical features of wear debris may induce high interaction of the surface debris with the surrounding body liquid phase, which could have, in case of the metal leaching, a significant impact on the serum metal levels [54].

### 3.3. Preferential Hydroxyapatite Layer Formation and Selective Metal Leaching Detected by High-Resolution SEM-EDS on the Same Wear Debris

To better understand the impact of the distinct wear debris nanotopographies observed by HIM (Figure 4B) on the biological environment and to gather further chemical information of the debris and surrounding tissue, the same specimen was transferred to SEM-EDS (Figure 5). The elemental mapping across the field-of-view of the distinct nanotopographies also identified by SEM (Figure 5A) revealed significant differences in some elemental concentrations between these regions as presented in (Figure 5B–D). Concentrations of Ca and P were found to be elevated on the sites with flat nanotopographies, indicating the preferential hydroxyapatite-like layer formation on the surface oxide layer, an effect just recently also observed in vitro [55]. The evolution and the kinetics of the hydroxyapatite-like formation on Ti alloys have been well known [56], where P is adsorbed to oxygen forming phosphate groups followed by Ca adsorption to eventually form calcium phosphate [57]. From the blue curve in Figure 5B, it is evident that the amount of P deposited on the Ti alloy surface is approximately two-fold higher on the flat region, the same accounts for Ca (see Table 1). Despite the investigated causal relation between oxygenated Ti surface and phosphate adsorption [58], oxygen was not found to be the key factor for increased P and Ca adsorption (see the elemental maps on a wider field-of-view on Appendix A). SEM-EDS analysis also confirmed P- and Ca-rich regions coinciding with remaining Al regions indicating the importance of Al for hydroxyapatite-like layer formation (see Appendix A). Furthermore, as the differences in the morphology nicely correspond to the differences in the elemental adsorption, the topographical features on a nm scale could be causally related to the extent of P and Ca adsorption and thus to the biological activity.

Moreover, the EDS analysis revealed that a substantial part of the alloy components V and especially Al were leached out of the particles. Less than one-tenth of Al (0.4–0.6%) and approximately one-half of V (1–2%) remained in the measured debris sites (see Table 1). As observed in previous work performed from the same biopsy tissue [59], the concentrations of Al and V in tissue were also lower than expected, taking into account the initial composition of the Ti-Al-V alloy. The explanation could be a selective V and especially Al leaching from the debris into the body fluid and surrounding tissue, which may be due to lack of charge and could be specific for this type of Ti-Al-V alloy. Such intense leaching could have severe implications on the surrounding tissue and even broader biological environment, as the elements have been found to cause severe immune and toxic effects [60,61]. These results can thus confirm the optically detected oxidative stress on biological matter in the vicinity of micron-sized wear debris.

### 3.4. Confirmation of Strong Al and V Leaching from Wear Debris by Micro-PIXE

To better characterize Al and V leaching on a wider field-of-view with approximately 10 times better sensitivity, the tissue specimen was eventually transferred to the micro-beam endstation for performing micro-PIXE measurements. Elemental maps of the tissue using Micro-PIXE are shown in Figure 6. These maps were obtained using GeoPIXE 7.5 software [62] (CSIRO, Australia), which allows the univocal elemental characterization of the features observed with optical microscopy. The main elements constituting the alloy, Ti, V and Al, were identified throughout the tissue in the shape of heterogeneous debris of sizes ranging from 200 µm down to the µm lateral resolution limit.

A fairly uniform elemental distribution of the wear debris was measured across a large field-of-view, confirming the presence of wear debris in the large part of the periprosthetic tissue. Furthermore, the analysis of the central particle (Figure 6, marked with a red oval on Ti map), using GeoPIXE software, with the Ti-Al-V alloy as a matrix for fitting and quantification, confirmed the result obtained with SEM-EDS (see Table 1, right column), where both V and Al concentrations in the alloy were found lower than the native concentrations, expressed as wt%. The concentration of remaining Al was found to be a few times higher compared to EDS analysis. This may be due to the limited interaction volume of EDS (order of few µm3), which could introduce some bias in the quantification in the case of Al not being evenly distributed throughout the alloy, changing its concentration from the bulk towards the surface of the material [63]. Tissue maps also revealed different behaviour of the leached elements. While the concentration of V was found to be low in comparison with the native alloy composition, but distributed following the pattern of Ti, the Al concentration was close to the limit of detection over almost all the scanned area, not following the pattern of Ti. There were a few exceptional areas with higher Al concentration that corresponded with the areas of higher concentration of P and Ca, which again agrees with the EDS measurements. This could be due to a significant effect of Al on the apatite-like layer formation or, conversely, to the calcification effect for the retention of Al.

The quantified decreased values of alloy elements may be explained by a specific dynamic of preferential leaching from the wear microparticle surface into the surrounding body media, where Al leaching is much faster than V and Ti. This explains the high correlation of Ti and V in the elemental images, whereas Al distribution follows a different, more diluted, pattern. This preferential leaching and transport is specific to Al in the Ti-Al-V alloy. A high amount of Al in periproprosthetic tissue has recently observed accompanying the failure of ceramic components made of alumina (study not published yet).

To track the release of metallic ions from the implant into the body, the metal concentrations were also measured in the patient’s blood. The comparison of metal concentrations found in the local periprosthetic tissue and body fluids is challenging given that different mass transport of particles and different metabolisms play roles in the accumulation of metals in the tissue and further transfer into the blood or urine. A high concentration of metallic particles found in the blood was a good indication of the level of implant degradation. The concentrations of Ti and V in blood were found to be 10 and 40 times, respectively, the normal values established at 6 µg/L for Ti, 0.14 µg/L for V, and 6.1 µg/L for Al, while the Al concentration was found to be close to the normal value. The increase of ion concentration in blood, particularly V, corresponds to the extent of element leaching from the wear debris observed with SEM-EDS and micro-PIXE. Regarding Al, its relatively low concentration, found in soft tissue and serum, does not rule out the possibility of Al accumulation in the body. Al can be transported around the body via the bloodstream, bound to transferrin [64], leading to toxic effects including malignancies [65], or playing a role in central nervous system degenerative diseases [66]. Due to its intrinsic proximity to the bone tissue, Al could also accumulate in the bones as it was recently found that metal-specific biomaterials can be accumulated in the peri-implant bone tissue [67]. The accumulation of Al in bone has also been related to bone fractures accompanying specific diseases [68].

To sum up, the combination of complementary experimental techniques was found to be successful for a detailed micro to nanoscale characterization of wear debris and its impact on the surrounding microbiological environment. The revealed impact of the tissue oxidative stress exhibited through lipopigment accumulation in the nearby lysosomes can be linked to the metallic-associated adverse local tissue reaction, where the debris and the metal ion release can stimulate a reaction leading to necrosis of the surrounding tissues [9]. Our findings, using the novel combination of advanced imaging and analytical techniques, can explain the origin of chronic inflammation, which was indicated by prior histologic examination and are summarised in Table 2.

## 4. Conclusions

In this study, we have addressed the relevant and complex biomedical problem of modular neck hip implant rejection and related chronic inflammation. Using a combination of correlative microscopy and analytical techniques covering wide spatial scales ranging from millimetres down to nanometres, we have characterized structural and functional properties of the wear debris, as well as their impact on the surrounding periprosthetic tissue obtained at revision surgery from a fractured modular neck of a primary bi-modular stem THA. Using hybrid optical microspectroscopy for sub-µm identification of biological response to wear debris, superior depth-of-field HIM for nm resolution and surface sensitivity imaging of wear debris, SEM-EDS for elemental quantification of individual wear debris and highly penetrative sub-µg/g sensitive micro-PIXE for elemental quantification of wear debris on a larger scale, we have covered broad spatial phases for in-depth characterization and functional analysis of the investigated system. By advanced label-free optical imaging, we have identified the oxidative effect of wear debris on the surrounding tissue by showing lipopigments accumulation in the nearby lysosomes. By HIM, using a correlative microscopy approach, we have revealed distinct wear debris surface topographies packed in the tissue showing severe fretting and corrosion and preferentially elevated hydroxyapatite-like layer formation corresponding to nanoscale features. By applying SEM-EDS elemental analysis on the same wear debris, we revealed a significant decrease in V, and particularly Al, concentrations, indicating their partial leaching in the tissue and body fluid. This is very likely responsible for the nearby biological response and confined hydroxyapatite-like layer formation at the debris sites, with the remaining high concentration of Al indicating the localized chemical activity. Finally, by micro-PIXE with higher elemental sensitivity, we observed a homogenous and wide spread of Ti alloy debris throughout the whole tissue sample and confirmed selective metal leaching that corresponds to elevated concentrations in the patient’s serum. The applied combination of the advanced microscopy and spectroscopy techniques can easily be adapted to any of the relevant clinical cases related to ALTR, as it can reveal more insights into implant rejection processes compared to the conventional histological examination further down on a submicron to single molecular scale.

## Figures and Tables

**Figure 1 materials-14-03048-f001:**
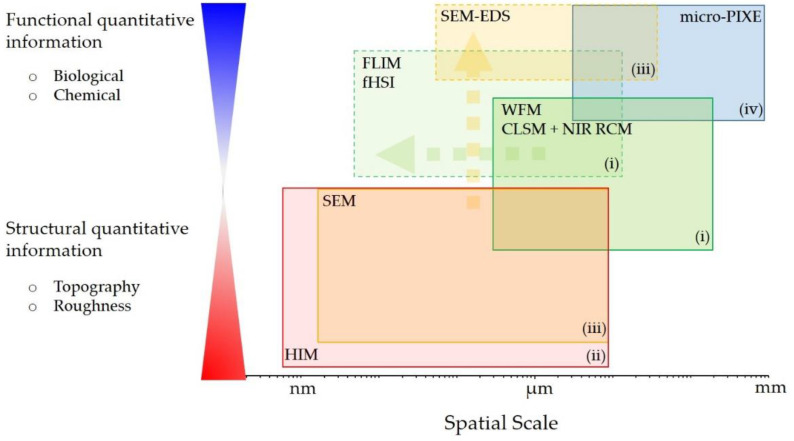
The experimental workflow phase diagram comprising complementary imaging and spectroscopy techniques (**i**–**iv**) to cover wide spatial range and to gain both functional and structural information on the investigated system. For more detailed instrumentation see Appendix A.

**Figure 2 materials-14-03048-f002:**
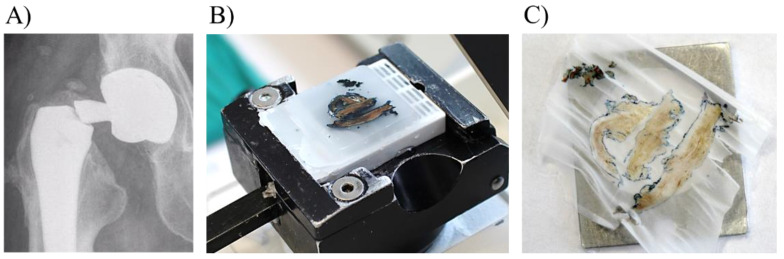
Collection and preparation of periprosthetic tissue slices for multimodal and multiscale analysis. (**A**) X-ray image from fractured modular neck (**B**) paraffin embedded periprosthetic tissue placed in the microtome (**C**) microtome sectioned tissue slice over the aluminium frame with mylar window.

**Figure 3 materials-14-03048-f003:**
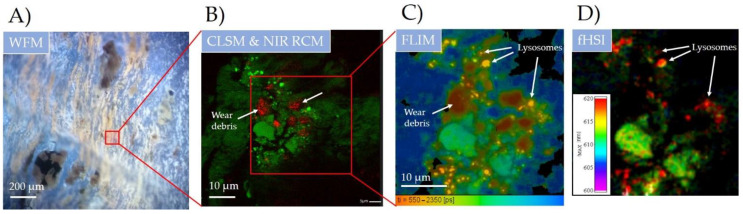
Wear debris identification and its impact on the surrounding biological molecular environment in the periprosthetic tissue on submicron scale using multimodal spectroscopy-based optical imaging. (**A**) widefield fluorescence microscopy (WFM) using LED source (λ = 400 nm); (**B**) hybrid LSCM and NIR CRM (λ = 561 nm and λ = 780 nm); (**C**) fluorescence lifetime (FLIM) and (**D**) hyperspectral fluorescence imaging (fHSI) using 16-channel PMT detector and fast TCSPC electronics (both λ = 561 nm). For more detailed FLIM and fHSI analysis see Appendix A.

**Figure 4 materials-14-03048-f004:**
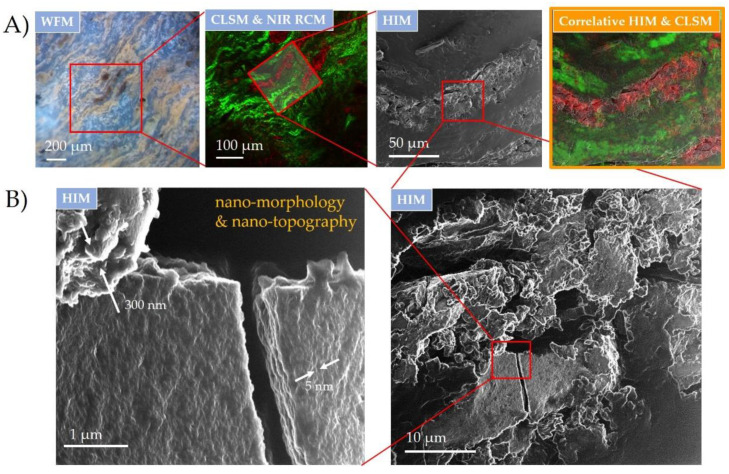
Surface topography of inflammatory wear debris distributed in periprosthetic tissue characterized down to nm scale using correlative HIM and optical CLSM and NIR RCM. (**A**) CM registration and overlap of wear debris site in the tissue sample on a micron scale; (**B**) wear debris surface nanotopography using HIM with distinct features from a few nm to a few hundred nm in size indicated with the white arrows. Excitation sources: WFM (LED; λ = 400 nm); CLSM and NIR RCM (λ = 561 nm and λ = 765 nm) and HIM SE (30 keV He ion energy with 1.7 pA ion current and 2 nm pixel step size).

**Figure 5 materials-14-03048-f005:**
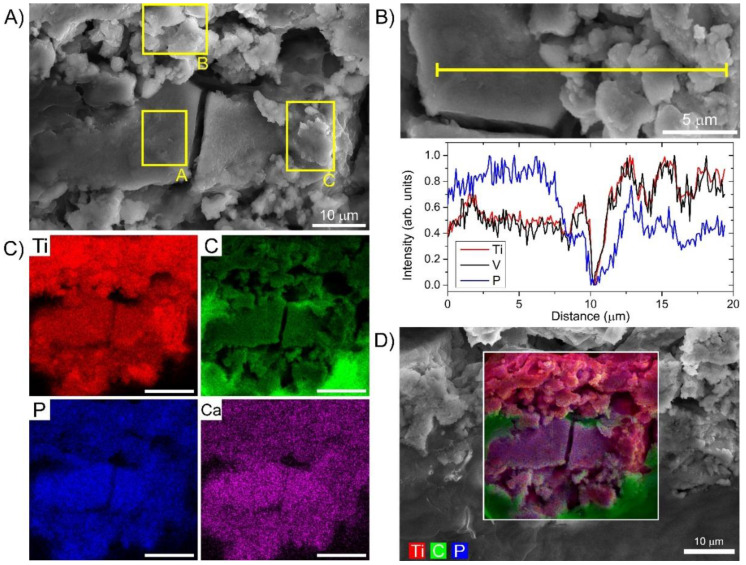
SEM-EDS analysis of Ti alloy wear debris with different local topographical features. (**A**) SEM image with the marked three distinct topographical regions; (**B**) SEM-EDS cross-section of Ti, V and P elemental concentration through these regions; (**C**) SEM-EDS overlay of three abundant elements, Ti from the alloy, P from the adsorption and C from the surrounding tissue; (**D**) EDS maps for four abundant elements, C, P, Ca and Ti.

**Figure 6 materials-14-03048-f006:**
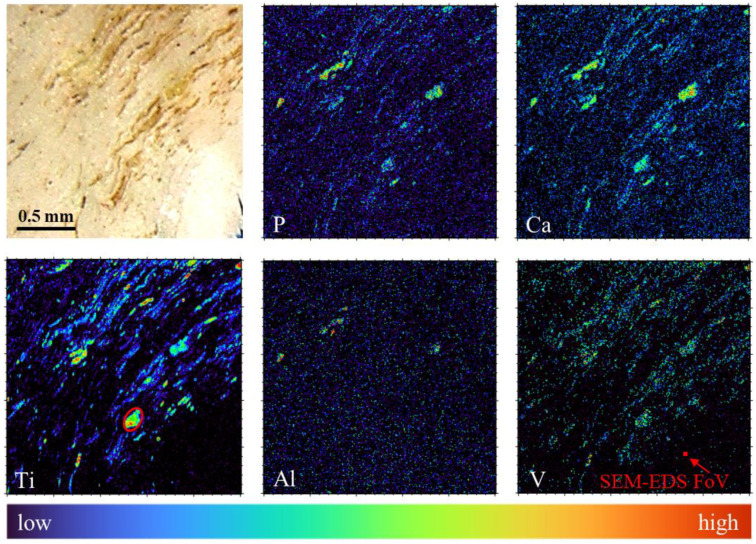
Qualitative elemental characterization of wear debris in periprosthetic tissue using micro-PIXE. Stereoscopic microphotograph (first image) and micro-PIXE elemental maps of P, Ca, Ti, Al and V in the scanned area (2 × 2 mm^2^). Red oval inserted in the Ti elemental map shows the particle selected for quantitative analysis. Red filled square in the V elemental map shows 30 × 30 µm^2^ field-of-view of the SEM-EDS analysis.

**Table 1 materials-14-03048-t001:** Total and normalized weight percent (wt%) of Ti_6_Al_4_V alloy debris and adsorbed elements on the marked distinct topographical regions. Initial wt% of the alloy components were 90% Ti, 6% Al and 4% V. For the elemental analysis of the same sample site on the wider FoV using SEM-EDS, see the Appendix A.

	SEM-EDS			Micro-PIXE
Elements\Site	Flat Region A	Rough Region B	Rough Region C	Larger Debris
Analysed area (µm^2^)	90	120	110	16,000
Ti [wt% _Total_]	7.0 ± 0.02	8.0 ± 0.02	9.0 ± 0.02	14.3 ± 0.2
**Ti [wt% _Norm-Alloy_]**	**98.3 ± 0.3%**	**98.0 ± 0.2%**	**97.6 ± 0.2%**	**97.3 ± 1.4%**
Al [wt% _Total_]	0.04 ± 0.01	0.03 ± 0.01	0.04 ± 0.01	0.19 ± 0.1
**Al [wt% _Norm-Alloy_]**	**0.6 ± 0.2%**	**0.4 ± 0.2%**	**0.4 ± 0.1%**	**1.3 ± 0.7%**
V [wt% _Total_]	0.08 ± 0.01	0.13 ± 0.01	0.18 ± 0.01	0.2 ± 0.1
**V [wt% _Norm-Alloy_]**	**1.1 ± 0.2%**	**1.6 ± 0.1%**	**2.0 ± 0.2%**	**1.4 ± 0.7%**
**P [wt% _Total_]**	**0.8 ± 0.01**	0.42 ± 0.01	0.48 ± 0.01	2.0 ± 0.1
**Ca [wt% _Total_]**	**0.66 ± 0.01**	0.33 ± 0.01	0.41 ± 0.01	0.87 ± 0.05
O [wt% _Total_]	7.2 ± 0.03	5.5 ± 0.03	6.7 ± 0.03	-
TOTAL	16.6	15	17.4	17.6 ± 0.6

**Table 2 materials-14-03048-t002:** Overview of the key findings of the wear debris in the periprosthetic tissue by advanced complementary experimental techniques.

Technique	Lateral Resolution	Technique Capability	Biomedical Observation
CLSM, NIR RCM	250 nm	Wear debris inside the tissue can be directly detected by NIR RCM where their sizes vary from sub µm to >10 µm.	Elevated autofluorescence was observed in the direct surrounding of the wear debris indicating on the biological response.
fHSI, FLIM	250 nm (*few nm—distinguish nearby molecular environment)	Wear debris was detected through the decrease in FLIM decay through the surface plasmon effect, [43] made possible by the presence of the lipopigment molecules in the direct vicinity.	Broad spectral width, the measured peak wavelength and the µm sized “granules” confirmed lipofuscin presence accumulated in lysosomes as a consequence of the biological oxidative stress response [69] which is induced by the nearby, micron-sized wear debris.
HIM	nm	HIM imaging showed high surface sensitivity with nm resolution that can easily differ between wear debris with different surface topographies.	The wide extent of morphologies can be attributed to the substantial wear damage due to fretting and corrosion that can induce inflammatory effect on the surrounding biological environment [53].
SEM-EDS	Sub µm (EDS)	Elemental analysis was possible on individual micron sized wear debris and showed non-homogeneous P and Ca deposition on its surface with distinct topographical features on a nm scale;	A substantial leaching of the alloy constituents Al and V was measured in the individual wear debris; Regions with remaining Al coincide with regions rich in P and Ca indicating Al-dependent hydroxyapatite layer formation or that the calcification effect may play a role in the retention of Al.
Micro-PIXE	few µm	The wear debris characterization was possible on a wider field-of-view with approximately 10 times better sensitivity and with the quantitative data matching SEM-EDS analysis.	Debris from Ti alloy is spread throughout the whole tissue slice; Elemental mapping confirmed significant leaching of Al and V from wear debris into the body during the implant corrosion process; Analysis confirmed the EDS finding of Al regions on wear debris coinciding with hydroxyapatite-like layer formation; Observed V leaching could be the main cause of significantly elevated concentration of V in patient’s serum.

## Data Availability

The data presented in this study are available on request from the corresponding authors and are available in Rossendorf Data Repository (RODARE) under https://www.hzdr.de/publications/Publ-32582, accessed on 30 April 2021 (doi:10.14278/rodare.966), and, after publication, in https://zenodo.org/, accessed on 30 April 2021, under “799182” keyword.

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
