# Peer review of "Revealing Inflammatory Indications Induced by Titanium Alloy Wear Debris in Periprosthetic Tissue by Label-Free Correlative High-Resolution Ion, Electron and Optical Microspectroscopy"

_materials, 2021, doi:10.3390/ma14113048_

Round 1

Reviewer 1 Report

The authors presented a manuscript to study the metallic-associated adverse local tissue reactions (ALTR) and proposed a combination of several techniques to study at sub-micrometer level this phenomenon. I find the manuscript well-written and interesting, however, I have some comments that need to be addressed.

1) Figure 1 is the same of reference 57, it is just vertically flipped.

2) The caption of figure 5 is the same of figure 4.

3) The quality of Figure 6 is quite low, especially compared with the same kind of picture previously published by the authors.

4) The statement "Current state-of-the-art imaging techniques in clinics to gather structural information of the effect of the implants on the surrounding tissue and biochemical information of cytokines involved in the immune response rely on magnetic resonance imaging (MRI [16–18], computed tomography (CT, SPECT-CT [19]) and immunohistochemical labelling [10,20], respectively. However, the techniques have limited spatial resolution, MRI and CT of only 500 μ m, and thus cannot provide any structural and functional information of the debris effect on the surrounding tissue on a microscale." could be misleading. When ALTR is suspected ultrasound, computed tomography, and magnetic resonance imaging using a metal-artifact reduction sequencing protocol are gold-standard. I suggest the authors to specify what kind of characterization they are talking about pre-implant failure or post-implant failure. 

Reviewer 2 Report

In this study, authors recruited several wonderful imaging techniques and provided nice pictures. The material was worn-broken titanium alloy based implant. Considering that titanium implant has been broken state, localized inflammation around implant should be present. Inflammation is accompanied by elevated ROS production. Thus, authors' indirect observation of ROS elevation might be due to wear debris or due to broken implant. As authors stressed biological responses to debris, cellular assay would be helpful.

Line 258-261 The elevated fluorescence could be a consequence of the oxidative stress as a response to pathological conditions or toxic compound [37], in this case, the nearby wear debris, that cause lipid peroxidation and thus the production of lipofuscin, being accumulated within lysosomes [38]. -> This sentences are unclear. Authors never measured ROS level in the tissue. The presence of lipofuscin was confirmed by indirect method. Accordingly, these were just assumption. If there was TEM image of cells, it would be better. But these findings were used as confirmative evidence in the abstract. In line 27-28, authors stated that "Micron-sized wear debris was found as the  main cause of the tissue oxidative stress exhibited through lipopigments accumulation in the nearby lysosomes." Based on authors' findings, authors just can state that "elevated fluorescence in their study". 
